# Trends and factors of botanical dietary supplement use among US adults with COPD from 1999 to 2016

**Yanjie Sun[1], Rongying Wang[1], Weiwei Tang[2,3,4], Chao Li[5], Nan Huo[6]***

**1** Department of General Practice, The Second Hospital of Hebei Medical University, Shijiazhuang, Hebei Province, China, **2** School of Health Policy and Management, Nanjing Medical University, Nanjing, Jiangsu, China, **3** Center for Global Health, Nanjing Medical University, Nanjing, Jiangsu, China, **4** Institute of Healthy Jiangsu Development, Nanjing Medical University, Nanjing, Jiangsu, China, **5** Auburn University Harrison School of Pharmacy, Auburn, Alabama, United States of America, **6** Department of Health Sciences Research, Mayo Clinic, Rochester, Minnesota, United States of America

* huo.nan@mayo.edu

**Data Availability Statement:** All data files are available from the National Health and Nutrition Examination Survey database from CDC website.

## Abstract

### Background

The potential effects of botanical dietary supplements (BDS) in the treatment of asthma have been proposed. However, the evidence of the effects of BDS use among patients with COPD is limited. The objective of our study was to exam the trends and effects of BDS use among US adults with COPD.

### Methods

A serial, cross-sectional study was conducted by using data of the NHANSE from 1999 to 2016 (n = 53,348). COPD (n = 2,580) was identified by using self-reported diagnosis history, and any BDS use was identified in the past 30 days. The prevalence of BDS use was calculated for respondents with and without COPD. Simple linear regression models were applied to test for trends in BDS use. Multiple logistic regression models were used to identify the factors of BDS use and patients' clinical outcomes, including all-cause hospitalization and abnormal hemoglobin levels, associated with BDS use. The results were weighted to represent national estimates.

### Results

The prevalence of BDS use was greater among participants who had COPD compared to the referent group (mean 16.77% vs. 15.11%, p = 0.044). The trend of BDS use decreased from 1999 through 2016 in COPD group (p = 0.0023), but the trend remained stable in the non-COPD group (P>0.05). Results of multivariate logistic regression models showed that patients with higher family income (≥100% FPL) and graduated from college were more likely to use BDS compared with non-users. BDS use was associated with a lower likelihood of having all-cause hospitalization (adjusted odds ratio = 0.64; 95% CI 0.45–0.92) and abnormal levels of hemoglobin (adjusted odds ratio = 0.67; 95% CI 0.49–0.92) among adult COPD patients, after adjusting for covariates.

Link: https://wwwn.cdc.gov/nchs/nhanes/Default.
aspx.

**Funding:** The authors received no specific funding
for this work.

**Competing interests:** The authors have declared
that no competing interests exist.

## Conclusions

The overall use of BDS decreased during 1999 through 2016 for US adults with COPD. The
potential benefit of these supplemental medications use for clinical outcomes might exist for
adult COPD patients.

## Introduction

Chronic obstructive pulmonary disease (COPD) is a common group of chronic lung condi-
tions with characteristics that are progressive and irreversible airflow obstruction [1]. COPD is
a worldwide public health condition that usually onsets in middle-aged or elderly persons,
especially those with a long history of smoking. COPD includes chronic bronchitis (CB) and
emphysema. Based on the CDC report, more than 10% of the population older than 45 years
in the United States has been diagnosed with COPD [2]. As the third leading cause of death in
the United States, the mortality from COPD has increased during the past 30 years in both
men and women [3]. Due to the high prevalence of COPD, both direct and indirect COPD
related medical costs exceed $40 billion annually in the United States [4, 5].

As a chronic condition, there is no cure for COPD [1]. COPD treatment goals involve
symptom management, reduction or prevention of recurrent exacerbations, lung function
improvement, and enhancing quality of life by using a variety of medications such as β-ago-
nists, corticosteroids, and leukotriene modifiers. However, long-term use COPD medications,
especially corticosteroids, patients usually need to face the quality of life reduction that is
induced by side effects. Therefore, similar to other chronic diseases, an increasing number of
Americans pursue complementary and alternative medicine (CAM) to improve general health
or quality of life [6, 7]. Botanical dietary supplements (BDS), which is the most popular type of
CAM that has been used for a variety of chronic diseases management such as asthma [8, 9],
diabetes, rheumatoid arthritis, and cancer [10].

Hemoglobin (Hb) abnormalities are common in COPD patients [11–13], which include
anemia and polycythemia. Although both of them are encountered in COPD, the prevalence
of polycythemia is much lower than anemia [14], and the prevalence of anemia might be
reported up to 46% [15, 16]. Hemoglobin can be a prognostic factor of COPD patients because
it is related to patients health status such as nutrition, comorbid diseases, medication, tissue
oxygen supply, and systemic inflammation [11, 17–19]. Although evidence has been indicated
the potential benefits of BDS for those chronic diseases, especially COPD similar disease—
asthma, there has not been any nationally representative studies to understand the patterns of
BDS use and assess their impact on clinical outcomes and Hb level of patients with COPD in
real-world settings. Therefore, the main objectives of this study were to understand the trend
of BDS use among COPD patients in the United States from 1999 to 2016 and to examine the
associations between BDS use and COPD related clinical outcomes by using National Health
and Nutrition Examination Survey (NHANES) data.

## Methods

### Data source and study population

This was a serial, retrospective cross-sectional study design by using 9 cycles of the NHANSE
datasets from 1999 to 2016. The NHANES dataset is a nationally representative cross-sectional
survey data that is conducted by the National Center for Health Statistics. It is a large-scale

continuing probability survey of households' representatives of the civilian non-institutionalized population in the U.S. It is conducted annually by the National Center for Health Statistics (NCHS) from the Centers at Disease Control and Prevention (CDC). In this study, adult participants (aged ≥18 years old) who completed continuous 2-year cycles from 1999 to 2016 household interviews were included. The overall unweighted response rate for the entire 9 cycles was 76%, and each specific cycle unweighted response rate was 69% to 84%.

For this study, we used the Medical Conditions file to identify COPD status (yes/no) based on two survey questions before the 2011–2012 cycle year, "Has a doctor ever told you that you have emphysema?" and "Do you still have chronic bronchitis?". After 2012, COPD status was identified by the question of "Has a doctor ever told you that you have COPD?". Adult participants were classified as having current COPD by answering "yes" to either one of these two questions. We also excluded individuals with missing or incomplete self-reported data on COPD status or education level. The final study sample includes 2,580 adults (weighted n = 14,793,841) with COPD and 50,768 without COPD (weighted n = 232,741,893). This study was approved by the Hebei Medical University institutional review board.

## Measurements

BDS use was measured as a binary variable from the NHANES DS data files based on the question of "During the past 30 days, have you taken any dietary supplements?" and calculated the number of botanicals in the supplements by using the variable of DSDCNTB in the dataset. If respondents' answers were "yes" or DSDCNTB ≥1, their BDS use was identified as "Yes," otherwise, their BDS use was defined as "No." The BDS use in this study included both single ingredient and combination products. The BDS included in the survey can be found on the NHANES website.

COPD clinical outcomes were identified as binary variables of having Hospitalization (Yes/ no) and having a lower level of hemoglobin (Yes/no) among COPD patients. The question from the Hospital Utilization & Access to Care (HUQ_B) file was asked to the respondents to determine whether they had overnight hospitalization in the past 12 months. The question was, "During the past 12 months, were you a patient in a hospital overnight?" The level of hemoglobin was identified from the Complete Blood Count with 5-part Differential—Whole Blood (L25_B) file. If respondents' hemoglobin levels were less than 13.5 g/dL in a man or less than 12 g/dL in a woman, they were defined as low levels of hemoglobin.

The following covariates were identified from the Prescription Medications and Medical Conditions files and the Demographic Variables and Sample Weights files: sociodemographic variables included sex (male and female), race/ethnicity (non-Hispanic white, others), age (≤39, 40–64, ≥65), marital status (married, never married, widowed or divorced, Unknown), annual family income status based on the Federal Poverty Level (FPL) (<100% FPL, 100%-200% FPL, 200%-400% FPL, >400% FPL, unknown), employment status (unemployed/unknown, employed), education level (high school or lower, some college, bachelor or higher), and health insurance status (Yes/No). Comorbidity and prescriptions were identified by using the self-reported diagnosis of health conditions (in total, 25 other health conditions excluding COPD) in the past 12 months, and the self-reported number of unique prescription medications.

## Statistical analysis

First, we estimated the overall prevalence trends of BDS use among adults with and without COPD. A simple linear regression model with a count-independent variable for the temporal cycle (cycles 1–9) was used to test (2-sided) for trends across 9 cycles. Generalized linear

model by using the generalized linear model (GENMOD) procedure and log-link function was used to compare the difference of marginal mean difference (MMD) in the prevalence of BDS use within each NHANES cycle between participants with and without COPD.

Then we compared covariates between patients with COPD with and without any BDS use in COPD samples using Chi-square tests. Proportions of participants with any all-cause hospitalization and low levels of hemoglobin were also compared between patients with COPD with and without any BDS use in COPD samples. Proportion estimates were calculated by using NHANES weights, which were calibrated to the US census totals for sex, age, and race/ ethnicity of the US population by each year cycle.

Three multivariable logistic regression models were used to identify factors associated with any BDS use (Model 1), and test the associations between any BDS use with COPD outcomes (Model 2 –having Hospitalization; Model 3 –having a lower level of hemoglobin) among adults COPD patients. All multivariable models were adjusted for covariates that were statistically significant in the bivariable analysis at P<0.05. We further tested the significance of the interaction term of BDS use and family income in model 2 and model 3 because previous literature has indicated that individuals with higher income are more likely to use supplemental treatments [20–22]. Results from multivariable models were weighted to represent national adult patients with COPD. The parameter estimates from the models were presented as adjusted odds ratio (AOR) and their corresponding 95% confidence intervals (CI). All analyses were conducted using SAS version 9.4 (SAS Institute, Inc., Cary, NC) and tests of statistical significance were conducted at the two-tailed α-level of 0.05.

## Results

### Results from trend analyses

Overall, there was a statistically significant decrease in the prevalence of any BDS use among patients with COPD during 1999 through 2016 ($P_{trend}$ = 0.0023) (Fig 1). However, there was no statistically significant change in the prevalence of any BDS use among patients without COPD during 1999 through 2016. The prevalence of BDS use was higher among patients with COPD (range: 21.30%-12.59%; overall mean = 16.77%) than among respondents without COPD (range: 12.30%-17.74%; overall mean = 15.51%) in each NHANES cycle.

### Results of factors associated with any BDS use

The characteristics of the study sample are presented in Table 1. From 1999 through 2016, in total, 17.19% of US adults with COPD reported using any BDS in the past 30 days (Table 1). Adjusted results showed that among adult COPD patients, BDS users tend to be non-Hispanic white, having higher family income, having health insurance, higher education levels, non-smokers, receiving fewer prescription medications, and had fewer other health conditions (Table 1 all P<0.05). However, results from the multivariable logistic regression model only found that COPD patients who had higher family income and higher education levels were more likely to use BDS compared to their counterparts. (Table 1).

### Associations between BDS use and COPD outcomes

Compared to adult COPD patients without BDS use (Table 2), those adults who used BDS had a lower rate (16.87% vs. 24.01%, P = 0.0151) of all-cause hospitalization in the past 12 months and a lower rate (14.78% vs. 18.10%, P = 0.0117) of low hemoglobin level. Multivariable results confirmed that BDS use was independently associated with a lower likelihood of having asthma-related ED visits (AOR = 0.64, 95% CI = 0.45, 0.92) and with the lower likelihood of

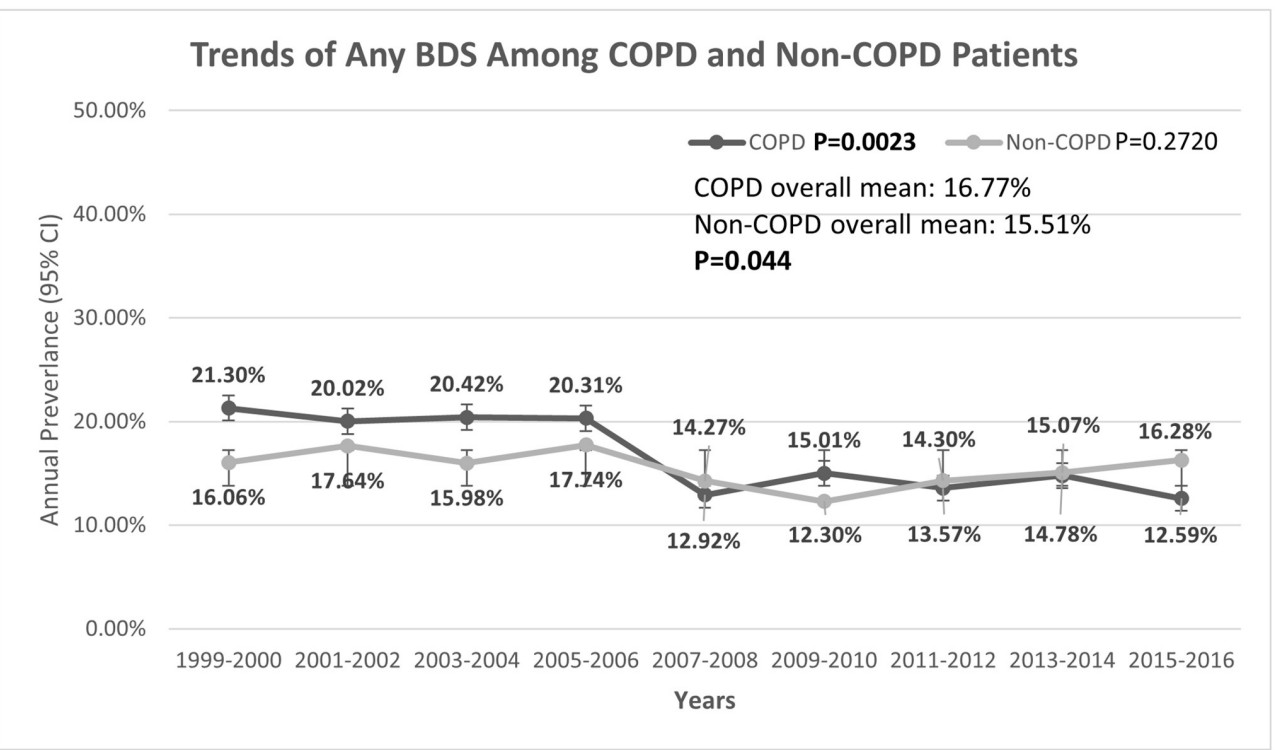

**Fig 1. Trends in the current use of any Botanical Dietary Supplements (BDS) among patients with COPD in the United States (weighted = 232,741,893).**

having low hemoglobin levels (AOR = 0.67, 95% CI = 0.49, 0.92), after adjusting for all covariates, among adult COPD patients. The interaction term of BDS use and family income was not significant (P = 0.149), indicating that income did not modify the association between BDS use and a lower likelihood of COPD related hospitalization and low hemoglobin level.

## Discussion

This study provides the first nationally representative profile of BDS use in adult populations with COPD. We found that about 17.19% of American adults with COPD used BDS from 1999 through 2016. The prevalence of BDS use among adult COPD patients was similar to its use in the general U.S. adult population (17.7%, 2012) [23]. However, after 2006, the prevalence of BDS use was lower than the general population, and more importantly, the trend of botanicals use among patients with COPD decreased from 21.3% to 12.59% through 1999 to 2016. Although our findings indicated a decreased trend of BDS use among COPD populations, the overall prevalence of BDS use among COPD patients was still slightly higher than non-COPD patients (16.77% vs. 15.51%). COPD is commonly diagnosed after the age of 45, and the highest prevalence is age above 65. Therefore, it is usually considered a disease of the elderly. Previous studies showed that, in some type of chronic diseases, such as cancer, and cardiovascular diseases, the trends of BDS were decreased, especially among elderly [10]. In addition, some studies indicated the patterns of all dietary supplements (DSs) among the general US population [24–26]. One possible explanation may be that several BDS have potential side effects, toxicity, and drug-drug interactions, For example, some plant extracts do have side effects [27–30], or even also have potential toxicity [31, 32]. In addition, some serious clinical

**Table 1. Sample characteristics for U.S. Adults with COPD and adjusted odds ratios of associations with Botanical Dietary Supplements (BDS) use among adult COPD patients (N = 2,580, weighted n = 103,484,187).**

| Variable | COPD Patients | | | | | Likelihood of Using BDS |
|---|---|---|---|---|---|---|
| | BDS Users N (%)[1] | | BDS Non-users N (%)[1] | | P* | AOR (95% CI)^ |
| | N = 374 | Weighted % (17.19) | N = 2206 | Weighted % (82.81) | | |
| **Age** | | | | | 0.3554 | |
| ≤39 | 78 | 25.56 | 437 | 23.30 | | |
| 40–64 | 167 | 50.88 | 971 | 49.05 | | |
| ≥65 | 129 | 23.56 | 798 | 27.65 | | |
| **Sex** | | | | | 0.2044 | |
| Male | 128 | 31.08 | 818 | 35.26 | | |
| Female | 246 | 68.92 | 1388 | 64.74 | | |
| **Race/Ethnicity** | | | | | **0.0136*** | |
| Non-Hispanic White | 265 | 84.69 | 1310 | 77.76 | | 1.42 (0.98–2.08) |
| Others | 109 | 15.31 | 896 | 22.24 | | ref |
| **Employment status** | | | | | 0.8555 | |
| Unemployed/unknown | 23 | 5.70 | 147 | 6.06 | | |
| Employed | 351 | 94.30 | 2059 | 93.94 | | |
| **Marital Status** | | | | | 0.7471 | |
| Never married | 33 | 9.86 | 274 | 12.37 | | |
| Married | 191 | 49.02 | 967 | 48.18 | | |
| Widowed or divorced | 116 | 28.36 | 680 | 26.35 | | |
| Unknown/inapplicable | 34 | 12.75 | 285 | 13.10 | | |
| **Annual Family Income** | | | | | **<.0001*** | |
| <100% FPL | 40 | 7.33 | 575 | 21.81 | | ref |
| 100–200% FPL | 102 | 22.69 | 661 | 26.77 | | **2.43 (1.54–3.84)** |
| 200–400% FPL | 105 | 30.73 | 476 | 26.02 | | **2.94 (1.69–5.12)** |
| >400% FPL | 105 | 32.24 | 306 | 19.20 | | **3.60 (2.15–6.05)** |
| Unknown | 22 | 7.01 | 188 | 6.20 | | |
| **Education Level** | | | | | **<.0001*** | |
| High school | 137 | 31.84 | 1294 | 53.10 | | ref |
| Some college | 141 | 39.75 | 630 | 30.81 | | **2.03 (1.49–2.77)** |
| Bachelor's degree | 96 | 28.41 | 282 | 16.09 | | **2.22 (1.41–3.49)** |
| **Health insurance** | | | | | **0.0004*** | |
| No | 52 | 14.60 | 351 | 16.11 | | ref |
| Yes | 322 | 85.40 | 1855 | 83.89 | | 0.89 (0.59–1.35) |
| **Smoking** | | | | | **0.0428*** | |
| Non-Smoker | 146 | 38.44 | 717 | 32.44 | | ref |
| Current Smoker | 228 | 61.56 | 1489 | 67.56 | | 0.94 (0.70–1.27) |
| **No. of prescription medications** | | | | | **0.0094*** | |
| **0** | 113 | 28.83 | 566 | 26.66 | | ref |
| **1–3** | 111 | 34.31 | 582 | 27.61 | | 1.12 (0.78–1.59) |
| **4–6** | 78 | 22.11 | 485 | 22.43 | | 0.93 (0.60–1.43) |
| **≥7** | 72 | 14.75 | 573 | 23.30 | | 0.66 (0.40–1.09) |
| **Other health conditions*** | | | | | **0.0025*** | |
| **0** | 51 | 16.26 | 231 | 11.23 | | ref |
| **1–2** | 185 | 52.55 | 1007 | 48.61 | | 0.76 (0.53–1.11) |

(*Continued*)

**Table 1.** (Continued)

| Variable | COPD Patients | | | | P* | Likelihood of Using BDS |
|---|---|---|---|---|---|---|
| | BDS Users N (%)[1] | | BDS Non-users N (%)[1] | | | AOR (95% CI)^ |
| | N = 374 | Weighted % (17.19) | N = 2206 | Weighted % (82.81) | | |
| ≥3 | 138 | 31.19 | 968 | 40.16 | | 0.69 (0.43–1.10) |

[1]Weighted percent

*Chi-square test significant at P<0.05

^All covariance variables that had Chi-square test P≥0.05 or small cell size (<20) were removed from the multivariable logistic regression models; adjusted odds ratios (AOR) and 95% confidence intervals (CI) were reported

* Other Health Conditions included 26 other health conditions excluding COPD

consequences may be caused by herb-drug interaction [33, 34], such as Salvia miltiorrhiza, Glycyrrhiza uralensis, P. ginseng and Angelica sinensis [35, 36].

The second goal of our study was to determine the specific factors associated with BDS use among adults with COPD in the US. Similar to our previous study of DSs use among US adults with asthma [8], we found that in adults with COPD, BDS users trend to be having higher income or having higher education. Our findings were not fully consistent with some previous studies. In our study, we did not find age, sex, race, smoking, and multiple health conditions were significantly associated with BDS use. Previous studies indicated that younger individuals or women were more likely to use non-vitamin dietary supplements in general population or some specific types of chronic diseases, such as asthma [8, 37, 38]. However, those previous studies either focused on the use of CAM and DSs among general populations or focused on patients with other chronic diseases, such as asthma instead of COPD. Comparing with asthma, COPD is more common in the elderly population, and adults with COPD usually have more comorbidities than adults with asthma, especially comorbid with cardiac diseases, such as heart failure [39]. Therefore, COPD patients may be more concerned about the potential side-effects of using BDS than asthma patients. In addition, our study found a positive association between higher levels of education and income with a greater likelihood of BDS use in adults with COPD, which is consistent with previous similar studies. Indeed, patients with chronic diseases, not only COPD, who have higher socioeconomic factors may be more likely to afford and purchase BDS for their overall health maintenance or improvement from

**Table 2. Associations between use of botanical dietary supplements with COPD clinical outcomes among adult COPD patients.**

| Variable | Adults COPD Patients [1] | | | | | | | | | |
|---|---|---|---|---|---|---|---|---|---|---|
| | Having hospitalization in last 12 months N (%)[1] | | | AOR (95% CI) | Low Hemoglobin N (%)[1] | | | | AOR (95% CI) |
| BDS Users | Yes | No | P* | | Yes | No | Unknown* | P* | |
| Yes | 65 (16.87) | 309 (83.13) | 0.0151 | 0.64 (0.45–0.92) | 67 (14.78) | 288 (80.99) | 19 (4.23) | 0.0117 | 0.67 (0.49–0.92) |
| No | 601 (24.01) | 1605 (75.99) | | ref | 503 (18.10) | 1490 (73.50) | 213 (8.40) | | ref |

^Other covariance variables had Chi-square less than 0.05 were collected into the multivariable logistic regression models; adjusted odds ratios (AOR) and 95% confidence intervals (CI) were reported

[1]Weighted percent.

*Chi-square

*Covariance variables in both hospitalizations and Hemoglobin level included sex, race, age, region, marital status, family income, prescription number, other health conditions.

*Hemoglobin level with missing were excluded from the multivariable logistic regression models.

physical and psychological aspects [6]. Finally, evidence has shown that DSs users are more likely to be non-smokers and less likely to have risk factors of chronic conditions such as cardiovascular diseases [40]. However, in our study, we did not observe those factors associated with BDS use. Overall, in current study, we only observed socioeconomic factors (income and education) were associated with BDS use among adults with COPD. Actually, socioeconomic factors may associates with many other possible factors such as better health behaviors, better diseases management [41], or response to conventional treatment, emotional support [42], as well as better health outcomes. However, due to the limitation of NHANSE data, some of socioeconomic factors cannot be controlled. In future study, more socioeconomic factors should be included to reduce potential confounding bias.

The third goal of this study was to examine the associations between BDS use and COPD-related clinical outcomes in adult populations with COPD. We found that the use of these supplement products among adults with COPD was associated with less likelihood (36% less) of self-reported all-cause hospitalization and low levels of hemoglobin (33% less) compared with those who did not use BDS. This is the first evidence that we are aware of to document the association between BDS use and COPD clinical outcomes using real-world, nationally representative population-level data. Although previous research has indicated potential health benefits of dietary supplements to COPD, such as improving lung function or FEV1/FVC, there are still several limitations in those studies such as small sample size or focusing on a few botanical products [43, 44]. A few studies indicated several BDS had a potential function to improve pulmonary function, to relieve symptoms, or to reduce exacerbation severity and frequency in the treatment of COPD based on clinical trials such as P. ginseng [45], H. helix, S. miltiorrhiza [43], and some traditional Chinese herbal medicine (TCHM). For example, in 2002, a well-designed clinical study indicated that treatment with Asian herb might be better relief of COPD symptoms than placebo. In that study, people who took ginseng experienced significant improvements in lung function and exercise capacity, compared to those who did not use [45].

Our findings also indicate that BDS use might improve low hemoglobin level in COPD patients. Anemia is a common comorbidity with the prevalence of in COPD ranges from 7.5% to 32.7% [46], which is associated with increased morbidity and mortality [47–50]. To our knowledge, our results indicated that BDS had the potential function to improve anemia in COPD patients. Although there is no solid evidence to support BDS may improve anemia from the biomedical aspect, some studies observed Chinese herbal medicine may prevent the decline of hemoglobin among patients with non-small cell lung cancer [51]. In addition, evidence supported that adults who use BDS had were more likely to have better health behaviors, such as physical activities [8], and less likely to have risk factors of chronic conditions such as cardiovascular diseases.

## Limitations

This study has several limitations. First, the self-report nature of survey data usually introduces recall bias. The majority of variables, particularly self-reported hospitalizations, were derived from single survey questions, which may affect the precision of measurements and thereby introduce measurement bias. Second, potential measurement errors in BDS use patterns from NHANSE survey data might exist, such as underreporting. Disease severity of COPD cannot be directly determined from survey data, although the prescription of COPD medications can be a proxy measure for those BDS users. Third, our findings may be influenced by unmeasured confounders such as other socioeconomic factors, medication adherence, response to conventional treatment, or emotional support. Fourth, the retrospective, cross-sectional observational

nature of the study design limits causality inference to the relationships between BDS treatment and COPD outcomes. Last but not least, this study did not capture potential confounders, such as type of care (e.g. specialty), medication use (type and adherence), environmental exposures, and different levels of health literacy of COPD patients.

## Conclusion

In conclusion, our findings provide national estimates of trends in BDS among U.S. adult patients with and without COPD. The prevalence of BDS use is higher among adults with COPD than adults without COPD. However, the trend of BDS use among adults with COPD was decreasing by year. We found a lower likelihood of having all-cause hospitalization and low hemoglobin levels associated with BDS use among adult COPD patients. Practitioners should closely monitor the patterns of BDS among COPD patients to help balance its benefits and risks in patient wellness and clinical outcomes. Further studies of the relationships of BDS use to clinical COPD outcomes and patient's quality of life are warranted.

## Author Contributions

**Data curation:** Yanjie Sun.

**Formal analysis:** Yanjie Sun, Chao Li.

**Investigation:** Nan Huo.

**Methodology:** Nan Huo.

**Project administration:** Nan Huo.

**Writing – original draft:** Yanjie Sun.

**Writing – review & editing:** Rongying Wang, Weiwei Tang, Chao Li, Nan Huo.

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
