## [Decision Letter · Decision Letter 0]

25 Aug 2020

PONE-D-20-22304

Trends and Factors of Botanical Dietary Supplement Use Among US Adults with COPD From 1999 to 2016

PLOS ONE

Dear Dr. Huo,

Thank you for submitting your manuscript to PLOS ONE. After careful consideration, we feel that it has merit but does not fully meet PLOS ONE’s publication criteria as it currently stands. Therefore, we invite you to submit a revised version of the manuscript that addresses the points raised during the review process.

I have received the comments of the reviewers on your manuscript. The specific comments of the reviewers are included below. Please provide point by point response in your revised manuscript.

We look forward to receiving your revised manuscript.

Kind regards,

Muhammad Adrish

Academic Editor

PLOS ONE

Journal Requirements:

2. We noticed minor instances of text overlap with the following previous publication(s), which need to be addressed:

(1) https://www.sciencedirect.com/science/article/abs/pii/S2213219817307419?via%3Dihub

(2) https://acsjournals.onlinelibrary.wiley.com/doi/full/10.1002/cncr.31183

The text that needs to be addressed involves the Results section (lines 190-195)  and the Discussion section (lines 244-285).

In your revision please ensure you cite all your sources (including your own works), and quote or rephrase any duplicated text outside the methods section. Further consideration is dependent on these concerns being addressed.

Reviewers' comments:

Reviewer's Responses to Questions

**Comments to the Author**

1. Is the manuscript technically sound, and do the data support the conclusions?

Reviewer #1: Yes

2. Has the statistical analysis been performed appropriately and rigorously? 

Reviewer #1: Yes

3. Have the authors made all data underlying the findings in their manuscript fully available?

Reviewer #1: Yes

4. Is the manuscript presented in an intelligible fashion and written in standard English?

Reviewer #1: Yes

5. Review Comments to the Author

Reviewer #1: The authors examine the use of supplements in COPD patients from NHANES data, over 18 years. They find a decreasing use over time and some demographic predictors of use.

Major comments:

It would be useful for the authors to note an example of the supplements being discussed

The authors have to be very careful in this cross sectional analysis to NOT imply causality ( i.e.- supplement users had fewer hospitalizations and and less anemia, but you can't say anything about directionality of this association)

SES factors were the strongest predictors of use ( and these are also strong predictors of health outcomes in the US) - thus- what was seen might still be a function of unmeasured SES confounders ( even in the multivariate analysis)

6. PLOS authors have the option to publish the peer review history of their article (what does this mean?). If published, this will include your full peer review and any attached files.

Reviewer #1: **Yes: **David Mannino

---

## [Author Response · Author response to Decision Letter 0]

1 Sep 2020

EDITOR'S SPECIFIC COMMENTS:

Thanks. We have made changes based on PLOS ONE’s style.

2. We noticed minor instances of text overlap with the following previous publication(s), which need to be addressed:

(1) https://www.sciencedirect.com/science/article/abs/pii/S2213219817307419?via%3Dihub

(2) https://acsjournals.onlinelibrary.wiley.com/doi/full/10.1002/cncr.31183

The text that needs to be addressed involves the Results section (lines 190-195) and the Discussion section (lines 244-285).

In your revision please ensure you cite all your sources (including your own works), and quote or rephrase any duplicated text outside the methods section. Further consideration is dependent on these concerns being addressed.

Thanks. We have rewritten these two parts and cited qualified references.

COMMENTS FROM REVIEWER #1:

1. It would be useful for the authors to note an example of the supplements being discussed

Thanks for this comment. We discussed one specific herb use in COPD management in the third paragraph of discussion.

2. The authors have to be very careful in this cross sectional analysis to NOT imply causality ( i.e.- supplement users had fewer hospitalizations and less anemia, but you can't say anything about directionality of this association)

Thank you for this comment. We have tried our best to avoid such statements, and in limitation part, we have discussed such causality issues of cross-sectional study.

3. SES factors were the strongest predictors of use ( and these are also strong predictors of health outcomes in the US) - thus- what was seen might still be a function of unmeasured SES confounders ( even in the multivariate analysis)

Thank you for pointing out SES factors. Yes, you are right, SES factors may be the strongest predictors of BDS use. However, due to the limitations of NHANSE data, not all SES factors can be observed or controlled. We have discussed more about this issue in the discussion and limitation parts.

---

## [Decision Letter · Decision Letter 1]

11 Sep 2020

Trends and Factors of Botanical Dietary Supplement Use Among US Adults with COPD From 1999 to 2016

PONE-D-20-22304R1

Dear Dr. Huo,

We’re pleased to inform you that your manuscript has been judged scientifically suitable for publication and will be formally accepted for publication once it meets all outstanding technical requirements.

Kind regards,

Muhammad Adrish

Academic Editor

PLOS ONE

Additional Editor Comments (optional):

Reviewers' comments:

Reviewer's Responses to Questions

**Comments to the Author**

1. If the authors have adequately addressed your comments raised in a previous round of review and you feel that this manuscript is now acceptable for publication, you may indicate that here to bypass the “Comments to the Author” section, enter your conflict of interest statement in the “Confidential to Editor” section, and submit your "Accept" recommendation.

Reviewer #1: All comments have been addressed

2. Is the manuscript technically sound, and do the data support the conclusions?

Reviewer #1: Yes

3. Has the statistical analysis been performed appropriately and rigorously? 

Reviewer #1: Yes

4. Have the authors made all data underlying the findings in their manuscript fully available?

Reviewer #1: Yes

5. Is the manuscript presented in an intelligible fashion and written in standard English?

Reviewer #1: Yes

6. Review Comments to the Author

Reviewer #1: (No Response)

7. PLOS authors have the option to publish the peer review history of their article (what does this mean?). If published, this will include your full peer review and any attached files.

Reviewer #1: **Yes: **David Mannino

---

## [Editor Report · Acceptance letter]

15 Sep 2020

PONE-D-20-22304R1

Trends and Factors of Botanical Dietary Supplement Use Among US Adults with COPD From 1999 to 2016

Dear Dr. Huo:

I'm pleased to inform you that your manuscript has been deemed suitable for publication in PLOS ONE. Congratulations! Your manuscript is now with our production department.

Kind regards,

on behalf of

Dr. Muhammad Adrish 

Academic Editor

PLOS ONE